# Selecting Multitarget Peptides for Alzheimer’s Disease

**DOI:** 10.3390/biom12101386

**Published:** 2022-09-27

**Authors:** Anne Kasus-Jacobi, Jennifer L. Washburn, Riley B. Laurence, H. Anne Pereira

**Affiliations:** 1Department of Pharmaceutical Sciences, University of Oklahoma Health Sciences Center, Oklahoma City, OK 73117, USA; 2Department of Physiology, University of Oklahoma Health Sciences Center, Oklahoma City, OK 73104, USA; 3Department of Pathology, University of Oklahoma Health Sciences Center, Oklahoma City, OK 73104, USA; 4Department of Cell Biology, University of Oklahoma Health Sciences Center, Oklahoma City, OK 73104, USA

**Keywords:** multitarget drug, neuroinflammation, neurodegeneration, cathepsin G, neutrophil, amyloid beta, receptor for advanced glycation end-products, Toll-like receptor 4, S100 calcium-binding protein A9, Alzheimer’s disease

## Abstract

Alzheimer’s disease (AD) is a multifactorial disease with a complex pathogenesis. Developing multitarget drugs could be a powerful strategy to impact the progressive loss of cognitive functions in this disease. The purpose of this study is to select a multitarget lead peptide candidate among a series of peptide variants derived from the neutrophil granule protein cathepsin G. We screened eight peptide candidates using the following criteria: (1) Inhibition and reversion of amyloid beta (Aβ) oligomers, quantified using an enzyme-linked immunosorbent assay (ELISA); (2) direct binding of peptide candidates to the human receptor for advanced glycation end-products (RAGE), the Toll-like receptor 4 (TLR4) and the S100 calcium-binding protein A9 (S100A9), quantified by ELISA; (3) protection against Aβ oligomer-induced neuronal cell death, using trypan blue to measure cell death in a murine neuronal cell line; (4) inhibition of TLR4 activation by S100A9, using a human TLR4 reporter cell line. We selected a 27-mer lead peptide that fulfilled these four criteria. This lead peptide is a privileged structure that displays inherent multitarget activity. This peptide is expected to significantly impact cognitive decline in mouse models of Alzheimer’s disease, by targeting both neuroinflammation and neurodegeneration.

## 1. Introduction

A role for neutrophils in the pathogenesis of Alzheimer’s disease (AD) has recently been proposed [1,2]. An increased number of neutrophils and a pro-inflammatory phenotype of these cells were shown, both in the circulation and in the brain parenchyma, in mouse models and patients with AD, compared to age-matched controls [3,4,5,6,7,8,9,10,11]. This indicates that peripheral inflammation involving neutrophils could play a role in AD, in addition to the central inflammation involving microglia that has been abundantly documented [12,13,14]. 

We previously reported a direct binding of three related neutrophil granule proteins to amyloid beta (Aβ), and to the receptor for advanced glycation end-products (RAGE), two important players in AD pathogenesis [2,15]. We hypothesized that these neutrophil proteins: cathepsin G (CG), neutrophil elastase (NE), and cationic antimicrobial protein of 37 kDa (CAP37) could modulate neurotoxicity in AD by influencing the Aβ-RAGE interaction [2]. In a follow up study, we found that NE and CG could cleave Aβ_1–42_, and inhibit its aggregation into oligomers and fibrils [16]. In contrast, CAP37 did not efficiently cleave Aβ_1–42_, but inhibited Aβ_1–42_ oligomerization and fibrillation, most likely through direct binding to the unaggregated form [16]. Of the three proteins, CG was found to be the most potent to inhibit the neurotoxicity of Aβ_1–42_, in a cultured neuron cell line [16]. In a different study, we looked for binding partners of CAP37 on live cells and found that CAP37 directly interacts with the Toll-like receptor 4 (TLR4) and with one of its natural ligands; the S100 calcium-binding protein A9 (S100A9) [17]. Furthermore, CAP37 was found to inhibit the activation of TLR4 by S100A9 [17]. The two other related neutrophil proteins, NE and CG, did not bind S100A9 or TLR4 [17]. These results, obtained in vitro using purified full-length proteins, are summarized in Figure 1.

Aβ, RAGE, S100A9, and TLR4 play important roles in AD pathogenesis by orchestrating the feed-forward mechanism between neuroinflammation and neurodegeneration, two pathways driving the cognitive decline in AD. Aβ, especially in the form of soluble oligomer aggregates, binds and activates a number of pattern recognition receptors (PRRs) in microglia, including RAGE and TLR4 [18,19,20]. This triggers phagocytosis of Aβ, release of pro-inflammatory cytokines and chemokines, and the release of S100A9 by these cells [19,21,22,23]. In turn, S100A9 binds and activates PRRs in neurons, including RAGE and TLR4 [24], which further increases the production of toxic Aβ [25]. Up-regulation of S100A9 was also found in neurons of patients with AD [25,26]. Upon neuronal cell death, it is released into the extracellular compartment and recognized as a damage-associated molecular pattern (DAMP) by PRRs in microglia [27]. In addition, S100A9 has intrinsic amyloidogenic properties, similar to those of Aβ, leading to oligomers, fibrils, and plaque formation, and has the ability to co-aggregate with Aβ, which further enhances neuroinflammation [25,28,29]. Neutralizing each of these targets (Aβ, RAGE, S100A9, and TLR4) individually has been shown to slow cognitive decline in animal models [30,31,32,33,34,35].

As previously described, synthetic ~25-mer peptides covering the entire CAP37 sequence have been tested to identify the domain binding to Aβ_1–42_ [16]. Peptide CAP37_120–146_, which is derived from the native sequence of CAP37 located between residues 120 and 146, was the highest binding peptide [16]. This sequence is shown in Figure 2. Furthermore, we found that the variant CAP37_120–146_ QH/WR, in which the native QH in positions 131 and 132 are changed to WR, inhibited oligomerization and neurotoxicity of Aβ_1–42_ [16]. In view of these results, we proposed a strategy to discover new drugs for AD, consisting of generating peptides that can mimic the beneficial effects of full-length neutrophil granule proteins [16]. In this study, we generated a new series of peptide variants, derived from the CG sequence located between residues 119 and 144, corresponding to the CAP37_120–146_ domain (see Figure 2). We chose CG based on the fact that CG was the most potent of the three full-length proteins to inhibit the neurotoxicity of Aβ_1–42_ [16].

This study was designed to screen these new peptide variants, and select a lead peptide to further test in vivo, based on the following criteria: (1) Inhibition and reversion of Aβ oligomers; (2) direct binding of peptide candidates to human RAGE, TLR4 and S100A9; (3) protection against Aβ oligomer-induced neuronal cell death; (4) inhibition of TLR4 activation by S100A9.

## 2. Materials and Methods

### 2.1. Materials

Custom peptides, derived from human CAP37 and CG native sequences (from residue 120 to 146 for CAP37 and 119 to 144 for CG), and human Aβ_1–42_ peptide were synthesized by CSBio (Menlo Park, CA, USA) with a purity ≥95%. Recombinant human calcium binding protein S100A9 (fused to a C-terminus 8-His tag) was purchased from Prospec Protein Specialists (Ness Ziona, Israel). Recombinant human TLR4/myeloid differentiation factor 2 (MD-2) complex (both proteins fused to a C-terminus 10-His tag) and recombinant human RAGE fused to the Fc portion of human IgG1 were obtained from R&D Systems (Minneapolis, MN, USA). The Oligomeric Amyloid-β (o-Aβ) ELISA Kit was from Biosensis (Thebarton, Australia). Mouse monoclonal anti-His tag primary antibody was from Abcam (Cambridge, MA, USA). Goat polyclonal anti-RAGE antibody was purchased from R&D Systems (Minneapolis, MN, USA). Donkey anti-mouse HRP-conjugated secondary antibody was from Jackson ImmunoResearch Laboratories (West Grove, PA, USA). Rabbit anti-goat HRP-conjugated secondary antibody was obtained from Pierce (Rockford, IL, USA). The mouse neuroblastoma cells Neuro-2a and 0.25% trypsin were from ATCC (Menassas, VA, USA). Phosphate buffered saline (PBS), Minimum Essential Medium α (MEMα), N2 Supplement, and Dulbecco’s Modified Eagle Medium without Phenol Red used for these cells were from Gibco (Waltham, MA, USA). The 10X PBS and penicillin/streptomycin were from Lonza (Basel, Switzerland). The human TLR4 reporter cell line HEKBlue, HEK Blue Selection, Normocin, and Quanti Blue were purchased from InvivoGen (San Diego, CA, USA). Fetal bovine serum was from Avantor Seradigm (Radner, PA, USA). Glutamax was from Gibco. Trypan blue (0.4%) and slides for TC 20 counter were from BioRad (Hercules, CA, USA). Tissue culture plates were from Falcon (Tewksbury, MA, USA). Low bind tips were from Sorenson (Salt Lake City, UT, USA), and low bind tubes were from Eppendorf (Enfield, CT, USA). Hexafluoroisopropanol (HFIP) and dimethyl sulfoxide (DMSO) were from Sigma Aldrich (St. Louis, MO, USA). Nuclease-free water was from Ambion (Austin, TX, USA). Sulfuric acid was from J.T. Baker Chemical Co. (Phillipsburg, NJ, USA). Citric acid and bovine serum albumin (BSA, # A3803) were purchased from Sigma Aldrich. O-Phenylenediamine (OPD) was from Kodak (Rochester, NY, USA). Maxisorp plates were from Nunc (Rochester, NY, USA). Other reagents were from Fisher (Pittsburgh, PA, USA).

### 2.2. Preparation of Aβ_1–42_ Oligomers

Ten mg Aβ_1–42_ was first dissolved in 2 mL of HFIP, incubated for 30 min at room temperature, aliquoted and allowed to dry in the fume hood overnight. Aliquots were then vacuum desiccated for 1 h at room temperature. Dry films of Aβ_1–42_ were stored at −20 °C to keep them in a monomeric form. Each aliquot was reconstituted in anhydrous DMSO at 5 mM before use by pipetting over the side of the tube, vortexing on high speed, and sonicating 5 min in a water bath sonicator. To form oligomers, HFIP-treated Aβ_1–42_ was diluted to 100 μM in MEM α and incubated overnight at 4 °C, in the absence or presence of 16 μM CAP37 or CG peptides. To determine whether Aβ_1–42_ oligomers could be reversed by CAP37 or CG peptides, oligomers were formed in the absence of peptide first, and then peptide candidates were added at 16 μM and incubated with oligomers for another 24 h at 4 °C.

### 2.3. Enzyme-Linked Immunosorbent Assays

Aβ_1–42_ oligomers, formed in the absence or presence of CAP37 or CG peptide candidates, were quantified using the Oligomeric Amyloid-β (o-Aβ) ELISA Kit from Biosensis, as described before [16]. Oligomers were formed/incubated in the absence or presence of peptide as described above. These preparations were then diluted (1/225,000 dilution) so that the concentration of monomeric Aβ equivalent falls within the linear range of detection of the kit. ELISAs were then performed as recommended by the manufacturer, and absorbance was read at 450 nm. To confirm direct interaction of CAP37 or CG peptides with binding partners, we conducted ELISA experiments, as described before [15,17]. Briefly, Nunc Maxisorp plates were coated with 5 μg/mL of BSA, CAP37-derived peptide, or CG-derived peptides prepared in PBS (pH 7.4), for 2 h at room temperature and then placed at 4 °C overnight. After coating, plates were washed with PBST (0.05% Tween 20 in PBS, *v*/*v*) and blocked with 3% BSA in PBST (*w*/*v)* for 1 h at room temperature. Next, plates were washed again, and His-tagged S100A9, His-tagged TLR4/MD-2, or RAGE-Fc chimera antigens, prepared in PBST 0.1% BSA, were added at 0 or 10 nM. Plates were incubated with antigens at 37 °C for 70 min, washed, and mouse monoclonal anti-His tag or anti-RAGE primary antibodies, prepared in PBST with 1% BSA, were added at concentrations of 0.5–1.0 μg/mL for anti-His tag, or 0.2–0.5 µg/mL for anti-RAGE. Plates were incubated at room temperature for 1 h and then washed. Next, the secondary antibody was prepared in PBST. Donkey anti-Mouse antibody was prepared at 0.08 μg/mL and rabbit anti-Goat antibody was prepared at 0.8 μg/mL, added to the plates and incubated at room temperature for 1 h. Following incubation with the secondary antibody, plates were washed, and 100 μL of OPD reagent prepared in citrate buffer (50 mM sodium citrate, 100 mM sodium phosphate, 0.8 mg/mL OPD, pH 5.0) was added to the wells and allowed to develop for 10 to 30 min in the dark. After development, reactions were stopped by adding 50 μL of 5-N sulfuric acid, and optical density (OD) values were recorded at 492 nm using a Synergy2 microplate reader and Gen5 1.11.5 software (BioTek Instruments, Inc., Winooski, VT, USA). 

### 2.4. Cell Culture and Neurotoxicity Assay

The mouse neuroblastoma cell line Neuro-2a (ATCC) was used for these experiments as previously described [16]. Cells were seeded in 96-well plates at 16,000 cells per well in MEM α with nucleosides and without phenol red, complemented with 10% heat inactivated fetal bovine serum and penicillin/streptomycin. At 24 h after seeding, cells were washed with warmed MEM α, and then incubated in 90 μL MEM α containing N2 Supplement + 20 μL of Aβ_1–42_ oligomer preparation, prepared in the absence or presence of peptide candidates. The final concentration of Aβ_1–42_ on the cells was 18 μM. Cells were incubated with Aβ_1–42_ treatments for 24 h prior to harvesting. Cells were then detached from the plate by pipetting up and down, mixed with trypan blue (0.2% final), and immediately counted in a BioRad TC20 cell counter. The percent dead cells were recorded in duplicates for each well.

### 2.5. Cell Culture and TLR4 Activation Assay

The HEK-Blue hTLR4 cells were used for these experiments as previously described [17]. Cells were cultured in complete growth medium: phenol red-free DMEM high glucose medium supplemented with 1 mM sodium pyruvate, 10% heat-inactivated FBS, 50 U/mL penicillin, 50 μg/mL streptomycin, 1X Glutamax, 100 μg/mL Normocin, and 1X HEK-Blue Selection. All stimulation assays were performed on cells that were passaged 3 to 15 times. Cells were carefully rinsed once with ice-cold PBS to remove residual FBS. Then, 2 mL of pre-warmed PBS was added, and dishes were incubated at 37 °C for 5 min to promote cell detachment. Cells were fully detached from plates with gentle tapping, collected, and centrifuged at 52× *g* for 5 min. Cell pellets were resuspended in complete growth medium without FBS at a final concentration of 1.5 × 10^5^ cells/mL. First, 180 μL of the cell suspension was added to the plate and allowed to settle for 2 h at 37 °C under 5% CO_2_. During this incubation, experimental treatments were prepared using protein LoBind tubes and low-binding tips. In one set of experiments aimed to determine activation of TLR4, treatments were prepared with increasing concentrations of S100A9 or peptides. The final concentrations of S100A9 were 0; 0.5; 2.5; 5; 25 and 50 nM. All peptides were tested at final concentrations of 0; 0.1; 1; 10; 100; 1000 and 10,000 nM. In another set of experiments, we determined the effects of peptide candidates on the activation of hTLR4 by S100A9. In this set of experiments, S100A9 was used at a constant final concentration of 1 or 5 nM, which leads to 75–80% of the maximum TLR4 stimulation. Treatments were prepared by mixing an equal volume of S100A9 with peptide and preincubating for 1 h on ice prior to addition to the cells. The final concentrations of peptides were 0; 1; 10; 100; 1000 and 10,000 nM. A positive control of 1 or 5 nM S100A9 alone was included in all stimulation assays. A total of 20 μL of experimental treatments were added to each well in triplicate, and the plate was incubated for 24 h at 37 °C under 5% CO_2_. Secreted alkaline phosphatase (SEAP) production was measured in each challenged well by transferring 20 μL of medium to a fresh 96-well plate containing 200 μL of Quanti-Blue detection medium. The *SEAP* reporter gene is under the control of an IL-12 p40 minimal promoter fused to five nuclear factor kappa B (NF-κB) and activator protein 1 (AP-1) binding sites. Stimulation of these cells by TLR4 ligands activates NF-κB, which in turn stimulates SEAP production. The Quanti-Blue plate was developed at 37 °C, 5% CO_2_, for 3 h followed by OD recordings at 630 nm using the Synergy2 microplate reader and Gen5 software. Mean OD values from untreated cells were subtracted as background from all other values. 

### 2.6. Statistical Analysis

Statistical analysis was performed using GraphPad Prism 9.3.0 (GraphPad Software, La Jolla, CA, USA). All statistical analyses were performed with a threshold for significance (alpha) set at 0.05. To analyze inhibition and reversion of Aβ oligomers by peptide candidates, we first subtracted the OD values obtained with peptide alone from the OD values obtained with Aβ ± peptide. We then used a one-way ANOVA, followed by a Kruskal–Wallis multiple comparisons test to compare the amount of Aβ oligomers formed/reversed in the absence and presence of peptide candidates. To analyze binding of antigens RAGE, S100A9 and TLR4/MD-2 to peptide candidates, we first subtracted the OD values obtained with 0 nM antigen from the OD values obtained with 10 nM antigen. We then used a one-way ANOVA and subsequent Dunnett’s multiple comparisons test to compare the binding of antigens to peptide candidates with binding to BSA, used as a negative control. For TLR4 activation analysis, we used a two-way ANOVA, followed by Dunnett’s multiple comparisons test. We compared the percent activation of hTLR4 by S100A9 alone with that of S100A9 combined with increasing concentrations of peptide candidates.

## 3. Results

### 3.1. Design of Peptide Candidates

Potentially important differences between the CAP37_120–146_ and CG_119–144_ sequences are: (1) the presence of two negatively charged residues in CG_119–144_ (D_137_ and E_141_) instead of two positively charged residues in CAP37_120–146_ in corresponding positions (R_136_ and R_142_); (2) an insertion of two residues (R_129_ and V_130_) in CG_119–144_ compared to CAP37_120–146_; and (3) a deletion of two residues (F_140_ and P_141_) in CG_119–144_ compared to CAP37_120–146_.

We show in Figure 2 the CG_119–144_ peptide variants tested in this study. These variants allowed us to test the effect of the negatively charged residues, insertion of RV, and deletion of FP on target engagement. Our comparator and positive control peptide variant, CAP37_120–146_ QH/WR, is also shown in Figure 2, next to the corresponding wild-type sequence of CAP37.

### 3.2. Effect of Peptide Candidates on Aβ_1–42_ Oligomerization

First, we formed Aβ_1–42_ oligomers in the absence and presence of each peptide candidate and quantified the number of Aβ_1–42_ oligomers formed in each reaction, as described in Materials and Methods. As shown in Figure 3A, the peptide CG (corresponding to the wild-type sequence of CG_119–144_) did not inhibit the formation of Aβ_1–42_ oligomers, while our positive control CAP37 QH/WR inhibited it by 80%, with a *p* value lower than 0.0001. In contrast, all CG variants had an inhibitory effect on Aβ_1–42_ oligomer formation. They all have in common the D/R substitution, suggesting that a positively charged residue instead of a negatively charged one in position 137 favors the binding of CG_119–144_ to Aβ_1–42_ monomers, thus preventing the formation of Aβ_1–42_ oligomers. The highest inhibition was mediated by peptides CG D/R, E/R (80% inhibition) and CG D/R, E/PR (90% inhibition), both with *p* values lower than 0.0001.

In another experiment, we first formed Aβ_1–42_ oligomers, and then incubated them in the absence and presence of each peptide candidate and quantified the amount of Aβ_1–42_ oligomers left at the end of the 24 h incubation. As shown in Figure 3B, the peptide CG did not significantly reverse (disaggregate) Aβ_1–42_ oligomers, while our positive control CAP37 QH/WR reversed more than 90% of Aβ_1–42_ oligomers, with a *p* value lower than 0.001. The D/R substitution in CG variants did not significantly change the results, suggesting that a positively charged residue in position 137 is not enough to induce oligomer reversion. However, when both negatively charged residues are changed to positively charged arginines, Aβ_1–42_ oligomer reversion was induced. The highest reversion was mediated by peptides CG D/R, E/R (>90% reversion, *p* < 0.0001) and CG D/R, E/PR (90% reversion, *p* < 0.01).

These results suggest that, at this point, our most promising peptide candidates are CAP37 QH/WR; CG D/R, E/R; and CG D/R, E/PR. The next logical step is to determine if these anti-aggregation effects translate into an inhibition of neurotoxicity mediated by the Aβ_1–42_ oligomers. 

### 3.3. Inhibition and Reversion of Aβ_1–42_ Oligomers Neurotoxicity by Peptide Candidates 

We first used Aβ_1–42_ oligomers formed in the absence and presence of peptide candidates to treat murine neuronal cells. As shown in Figure 4A, Aβ_1–42_ oligomers formed without peptide induce close to 60% of neuronal cell death in 24 h. All peptide candidates except CG partially inhibit the toxicity Aβ_1–42_ oligomers, when they were formed in the presence of these peptides. As shown in Figure 4A, these protective effects of peptides are only partial and do not always correlate with the quantified number of oligomers shown in Figure 3A. For example, CAP37 QH/WR inhibited the formation of oligomers by 80% but only inhibits neurotoxicity by less than 25%, with a *p* value of 0.0711. Absence of correlation between the two outcomes suggests that CAP37 QH/WR-disrupted oligomers were not recognized by the anti-oligomer antibody, but were still neurotoxic. CAP37 QH/WR could be masking the epitope recognized by the antibody in oligomers, without significantly decreasing their neurotoxicity. 

We then used Aβ_1–42_ oligomers formed in the absence of peptide and subsequently reversed by peptide candidates to treat the neurons. As shown in Figure 4B, results for reversion of toxicity are similar to results obtained for inhibition of toxicity, shown in Figure 4A. All peptide candidates except CG partially inhibit the toxicity of Aβ_1–42_ oligomers, after they were reversed in the presence of these peptides. Once again, these effects are partial and do not correlate well with the number of oligomers left after reversion, shown in Figure 3B. The only statistically significant result was obtained with peptide CG D/R, E/R, inducing 45% reversion with a *p* value of 0.0488. 

### 3.4. Interaction of Peptide Candidates with RAGE, S100A9 and TLR4

Direct binding of RAGE, S100A9, and TLR4 to each peptide candidate was quantified by ELISA, as described in Materials and Methods. As shown in Figure 5, the peptide CG did not bind RAGE (Figure 5A), S100A9 (Figure 5B), or TLR4 (Figure 5C). The D/R substitution allowed partial binding of RAGE and TLR4, but did not allow binding of S100A9. Additional E/R substitution increased binding of RAGE and TLR4 to levels seen with the positive control CAP37 QH/WR. The E/R substitution also allowed binding of CG peptide to S100A9. The highest binding of all three targets (RAGE, S100A9, TLR4) was found with peptide CG D/R, E/R. The peptide variant CG D/R, E/PR did not perform as well, and the RV deletion variants performed consistently worse than the corresponding undeleted variants. These results suggest that having the two negatively charged residues replaced by positively charged arginines is crucial to allow interaction of the CG peptide with its targets. Another important finding is that the RV insertion in CG_119–144_ is important as well for these interactions and should be kept in the CG peptide variants. Taken together, these results designate CG D/R, E/R as our top peptide candidate in this series of CG peptide variants.

### 3.5. Inhibitory Effects of Peptide Candidates on TLR4 Activation 

Because some of these peptide candidates appear to be ligands of RAGE and TLR4, two pro-inflammatory receptors involved in AD pathogenesis, it is important to rule out the possibility that they could activate these receptors as agonists. We used a TLR4 reporter cell line to quantify TLR4 activation in the presence of increasing concentrations of peptides. We used S100A9 as a positive control because it is a known agonist of TLR4. As shown in Figure 6A, we observed a dose-dependent activation of TLR4 by S100A9. Maximum receptor stimulation is obtained with 50 nM S100A9 (5 × 10^−8^ M). The half maximal effective concentration (EC_50_) of S100A9 is 0.7 nM (0.7 × 10^−9^ M). Along with S100A9, we tested four peptide candidates at concentrations up to 100 μM (10^−5^ M). As shown in Figure 6A, none of them was found to activate TLR4. 

Our next objective was to determine if any of these peptide candidates could modulate (inhibit or potentiate) the activation of TLR4 by S100A9. We used 1 or 5 nM of S100A9 and added increasing concentrations of peptide candidates to the cell treatments. As shown in Figure 6B, the top three peptide candidates inhibit the activation of TLR4 in a dose-dependent manner. We arbitrarily set the TLR4 activation by 1 or 5 nM S100A9 alone at 100% in this graph, to provide a direct visualization of the % inhibitory effects of peptides. The wild-type CG peptide, which does not interact with TLR4 or S100A9, does not significantly modulate TLR4 activation until the highest concentration. In contrast, positive control CAP37 QH/WR significantly inhibits TLR4 activation by up to 30%, and peptides CG D/R, E/PR and CG D/R, E/R significantly inhibit it by up to 60%, all three peptides with *p* < 0.0001. The inhibitory effects of peptides CAP37 QH/WR; CG D/R, E/PR; and CG D/R, E/R could be due to quenching of S100A9 because S100A9 was mixed and incubated with peptide first, before addition to the cells. It is also possible that any excess of peptide not bound to S100A9 could bind TLR4 and keep the receptor in an inactivated conformation, either by competing with S100A9 binding or through an allosteric modification of the receptor. 

## 4. Discussion

Results are compiled in Table 1. We did not include inhibition and reversion of Aβ_1–42_ oligomers in this table, but chose to include only inhibition and reversion of oligomers’ neurotoxicity, because the two outcomes (amount of oligomers and neurotoxicity) did not always correlate, and inhibition of neurotoxicity is the effect that we ultimately want to achieve. For each outcome included in Table 1, we used “+” and “–” signs, to indicate the level of activity of each peptide candidate. We used these signs in the following manner: ++++ for 75–100% of the highest effect observed; +++ for 50–75%; ++ for 25–50%; + for 5–25%; and – for close to 0 ± 5%. After looking at inhibition/reversion of neurotoxicity and binding of RAGE, S100A9, and TLR4, we added up the number of “+” signs for each peptide, and assigned the corresponding number of points to each candidate. Only the top 3 candidates, CAP37 QH/WR and CG D/R, E/PR with 14 points each, and CG D/R, E/R with 16 points, were tested for TLR4 inhibition. The CG peptide candidate had 0 points and was used as a negative control in these experiments. The other peptides were not determined (indicated by ND in Table 1). After including the TLR4 inhibition results, CG D/R, E/R was found to be the lead peptide candidate, with the highest number of final points (19 points). CG D/R, E/PR and CAP37 QH/WR are the next two candidates in line, with a lower number of 17 and 16 final points, respectively. These peptides are multitarget compounds.

Multitarget drugs can be of different types, and the ones with inherent multitarget activity such as our selected lead peptides are called privileged structures, as opposed to multitarget drugs designed by combining of two or more single-target domains together [36]. Because the pathogenesis of AD is complex and involves multiple pathways, combination drug therapy or multitarget drugs have the potential to impact the progression of AD significantly more than a single drug with only one target [36]. Selecting an appropriate single target to hit is an extremely difficult task, as suggested by the high number of clinical failures when testing single-target disease-modifying drugs for AD [35,36]. Among the multiple mechanisms that influence the progression of AD, such as accumulation of Aβ, chronic neuroinflammation, vascular dysfunction, formation of tau tangles, and neurodegeneration, neuroinflammation and neurodegeneration are particularly important [37]. These two pathways are triggered by the accumulation and aggregation of Aβ in the brain, they are concomitant, each pathway enhances the other in a feed-forward mechanism, and they are the main drivers of the cognitive decline in AD [2]. Therefore, a multitarget peptide such as CG D/R, E/R could lead to a synergistic effect on cognitive decline by targeting both neuroinflammation and neurodegeneration.

Our selected lead peptide CG D/R, E/R will advance to in vivo testing in mice. It will be tested and optimized for brain penetration, stability, and for engagement of the 4 targets (Aβ_1–42_ oligomers, RAGE, S100A9, and TLR4) in the mouse brain. Route of administration and formulation will be optimized, and we will conduct pre-clinical efficacy studies using mouse models of AD. Our hypothesis is that it will synergistically impact cognitive decline by targeting both neuroinflammation and neurodegeneration. We will use other CG peptides in this series, with different levels of engagement of each of the four targets, as controls to test this hypothesis in mouse models of AD.

## 5. Conclusions

This in vitro study allowed us to select a lead candidate among a series of peptide variants derived from the neutrophil granule protein cathepsin G. Our lead candidate CG D/R, E/R is a multitarget peptide that is expected to have synergistic effects on the cognitive decline associated with Alzheimer’s disease.

## 6. Patents

An International patent application (#PCT/US19/36281) was submitted in June 2019. A US Continuation-In-Part was filed on 12 August 2020, and protection is being pursued in Europe and Japan. Another international patent application (#PCT/US21/62308) was filed on 12 August 2020.

## Figures and Tables

**Figure 1 biomolecules-12-01386-f001:**
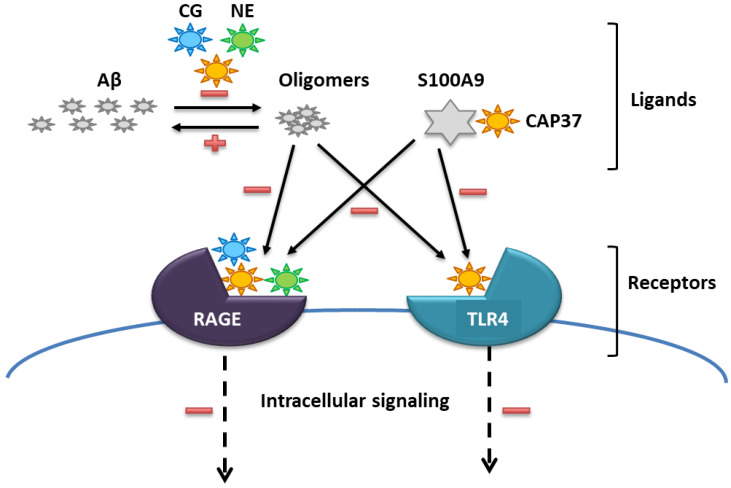
Effects of full-length neutrophil granule proteins on target proteins. Purified CG (shown in blue), CAP37 (shown in orange), and NE (shown in green) directly bind Aβ and RAGE [15]. These proteins inhibit (shown by the minus sign) and reverse (shown by the plus sign) the aggregation of Aβ_1–42_ into neurotoxic and proinflammatory oligomers [16]. They also disrupt the Aβ-RAGE interaction by binding to RAGE [15]. Additionally, CAP37 binds S100A9 and TLR4, disrupting the S100A9-TLR4 interaction and the activation of TLR4 by S100A9 [17].

**Figure 2 biomolecules-12-01386-f002:**
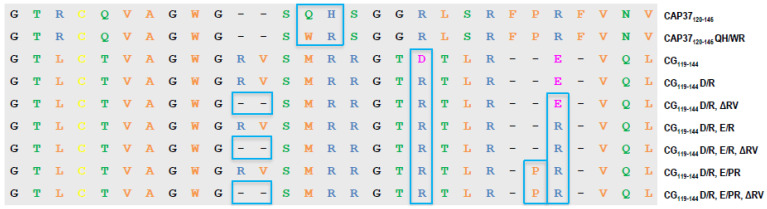
Sequence alignment of peptide variants derived from CAP37 and CG. The Aβ-binding domain of CAP37 (CAP37_120–146_) and corresponding sequence in human CG (CG_119–144_) are shown, along with the sequence of all peptide variants tested in this study. Negatively charged amino acids are indicated in pink, positively charged in blue, hydrophobic in orange, and hydrophilic in green. Glycine residues are shown in black, and cysteine in yellow.

**Figure 3 biomolecules-12-01386-f003:**
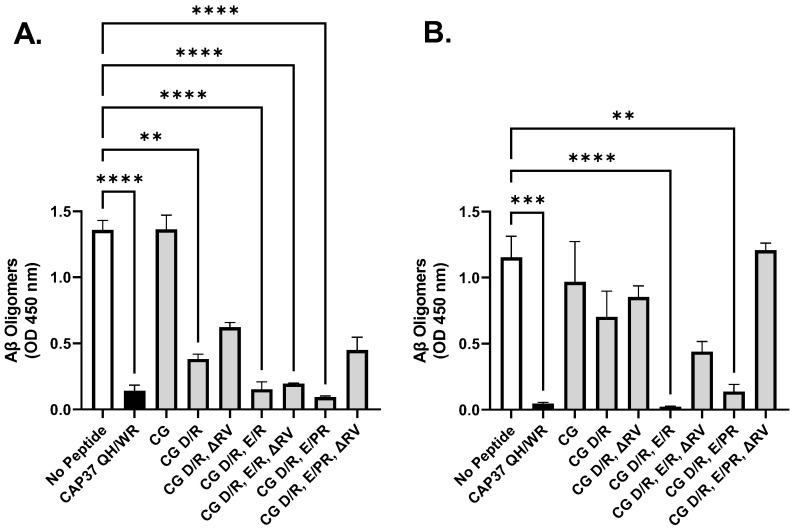
Effect of peptide candidates on Aβ_1–42_ oligomerization. (**A**) For inhibition of oligomerization, monomeric Aβ_1–42_ was prepared at 100 μM and incubated for 24 h to form oligomers, in the absence or presence of 16 μM of each peptide candidate. (**B**) For reversion of oligomers, they were prepared in the absence of peptide as described in (**A**). Then, 0 or 16 μM of each peptide candidate was added to the formed oligomers, and incubated for another 24 h. Aβ_1–42_ oligomers were quantified by ELISA, and bar graphs show average ± SEM of OD values, after background subtraction, reflecting the concentration of oligomers at the end of each reaction. Background value for each reaction was obtained by setting up a control with vehicle replacing Aβ_1–42_ in the reaction. At least two independent experiments were conducted, with each reaction performed in duplicate or triplicate. A Kruskal–Wallis multiple comparisons test was performed to compare the amount of Aβ_1–42_ oligomer formed/reversed in the absence and presence of each peptide candidate. Significant differences are shown as ** for *p <* 0.01, *** for *p <* 0.001, and **** for *p <* 0.0001.

**Figure 4 biomolecules-12-01386-f004:**
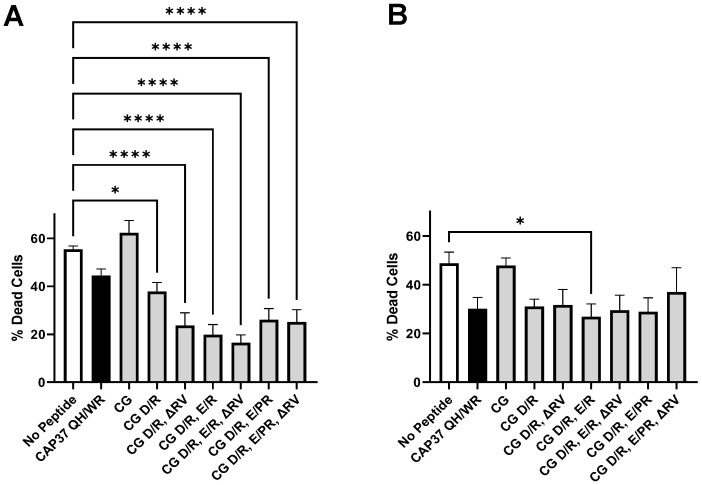
Effect of peptide candidates on Aβ_1–42_ oligomers’ neurotoxicity. (**A**) For inhibition of oligomers’ neurotoxicity, Aβ_1–42_ oligomers were prepared in the absence and presence of peptide candidates, as described in Figure 3A and added to mouse neuroblastoma cells at a final concentration of 18 μM. (**B**) For reversion of oligomers’ neurotoxicity, Aβ_1–42_ oligomers were prepared and then incubated in the absence and presence of peptide candidates, as described in Figure 3B and added to mouse neuroblastoma cells at a final concentration of 18 μM. After 24 h incubation with treatments, cells were detached, labeled with trypan blue, and counted. Bar graphs show the average of % dead cells ± SEM for each treatment, after background subtraction. Background was obtained by treating the cells with reactions in which Aβ_1–42_ was replaced with vehicle. Results are from three independent experiments, each performed in duplicate and counted twice. A Kruskal–Wallis multiple comparisons test was performed to compare the % cell death in the absence and presence of peptide candidates. Significant differences are shown as * for *p <* 0.05, and **** for *p <* 0.0001.

**Figure 5 biomolecules-12-01386-f005:**
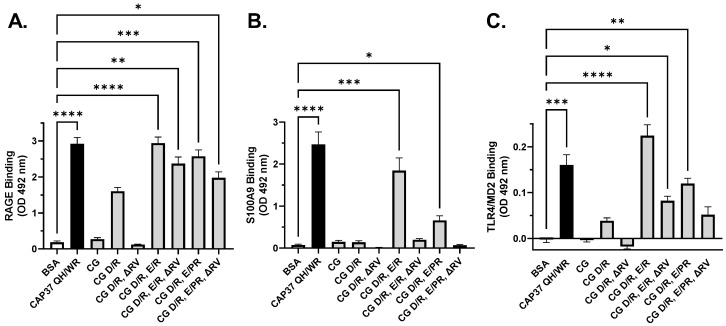
Binding of RAGE, S100A9, and TLR4/MD2 to peptide candidates. ELISA plates were coated with BSA or the indicated peptide candidates. RAGE (**A**), S100A9 (**B**) or TLR4/MD2 (**C**) antigens were added to coated wells at 0 or 10 nM. Bound antigens were quantified using specific antibodies. Bar graphs show the average ± SEM of OD values, after background subtraction. Values obtained with 0 nM antigen were subtracted as background from the corresponding values obtained with 10 nM antigen. A Kruskal–Wallis multiple comparisons test was performed to compare the binding of each antigen to peptide candidates with the binding of each antigen to BSA. Significant differences are shown as * for *p <* 0.05, ** for *p <* 0.01, *** for *p <* 0.001, and **** for *p <* 0.0001.

**Figure 6 biomolecules-12-01386-f006:**
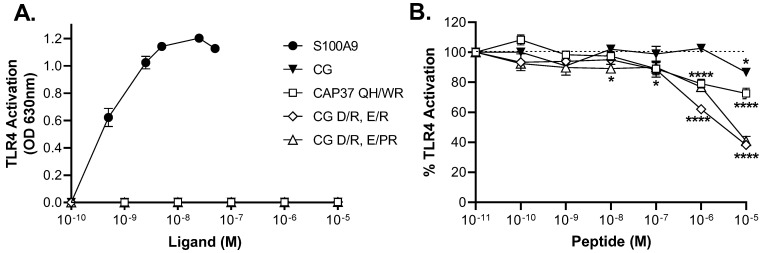
TLR4 activation by S100A9 and effects of peptide candidates. (**A**) SEAP production was measured in cell medium following 24 h incubation with indicated concentrations of ligands. (**B**) Inhibitory effect of peptide candidates on S100A9-induced activation of TLR4. TLR4 was activated with 1 or 5 nM S100A9 alone, or in combination with indicated concentrations of peptide candidates. Bar graph shows average % activation of TLR4 ± SEM relative to activation induced by 1 or 5 nM S100A9 alone, arbitrarily defined as 100%. Results are from three independent experiments, each performed in triplicate. A two-way ANOVA, followed by Dunnett’s multiple comparisons test was used to compare activation of TLR4 by S100A9 alone with activation of TLR4 by S100A9, in the presence of increasing concentrations of peptide candidates. Significant differences are shown as * for *p <* 0.05 and **** for *p <* 0.0001.

**Table 1 biomolecules-12-01386-t001:** Summary of lead peptide selection.

Peptide Candidate	Inhib. Neurotox.	Rev. Neurotox.	Bind RAGE	Bind S100A9	Bind TLR4	Points	Inhib. TLR4 Activation	Final Points
**CAP37 QH/WR**	**+**	**++**	**++++**	**++++**	**+++**	**14**	**++**	**16**
**CG**	**-**	**-**	**-**	**-**	**-**	**0**	**-**	**0**
**CG D/R**	**++**	**++**	**++**	**-**	**+**	**7**	**ND**	**ND**
**CG D/R ΔRV**	**+++**	**++**	**-**	**-**	**-**	**5**	**ND**	**ND**
**CG D/R E/R**	**+++**	**++**	**++++**	**+++**	**++++**	**16**	**+++**	**19**
**CG D/R E/R ΔRV**	**+++**	**++**	**++++**	**-**	**++**	**11**	**ND**	**ND**
**CG D/R E/PR**	**+++**	**++**	**++++**	**++**	**+++**	**14**	**+++**	**17**
**CG D/R E/PR ΔRV**	**+++**	**+**	**+++**	**-**	**++**	**9**	**ND**	**ND**

## Data Availability

Not applicable.

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
