# Peer review of "Selecting Multitarget Peptides for Alzheimer’s Disease"

_biomolecules, 2022, doi:10.3390/biom12101386_

Round 1

Reviewer 1 Report

The manuscript entitled "Selecting Multitarget Peptides for Alzheimer’s Disease" by  Kasus-Jacobi et al. is a nice and interesting study which adds notably to the existing literature. Overall the manuscript is well received and carries out a study of significance. However, there are some concerns that need to be cleared before it can deemed suitable for acceptance in the journal Biomolecules

1. Why did authors choose to stay exclusively to in vitro studies. Ideally it would have been wonderful to study the effect of these peptides in a suitable in vivo model.

2. I could not find any ethical statement. Was there any? If yes please share the body who issues the ethical approval.

Author Response

We would like to thank this reviewer for his/her positive comments on this manuscript. Please find the answers to your specific questions below:

Question 1: Why did authors choose to stay exclusively to in vitro studies. Ideally it would have been wonderful to study the effect of these peptides in a suitable in vivo model.

We simply did not have the resources to test all eight peptide variants in vivo. However, now that we have selected a lead peptide (based on this in vitro work) we can apply for more funding to test the lead peptide in vivo. During the course of our future in vivo studies, we may also test the effects of some of the other peptide candidates described in this manuscript.

Question 2:  I could not find any ethical statement. Was there any? If yes please share the body who issues the ethical approval.

Since this manuscript did not include any human or animal research, we did not include any ethical statement. However, we indicated the commercial sources of the two cell lines that were used in this work, as requested by the Journal under the "Research and Publication Ethics" section of the Instructions for Authors.

Reviewer 2 Report

Overall, this was a well-written and scientifically sound paper. I have no major revisions to suggest. The authors present the background and study design clearly. Results are adequately discussed, and figures are clear. I appreciated the table at the end to summarize all results. This paper will be of interest to anyone working on Alzheimer's disease. It presents an alternative to small molecule development for modulating this disease. I strongly recommend this for publication. 

Author Response

We would like to thank this reviewer for his/her positive comments on this manuscript and on the value of our work.

Reviewer 3 Report

Dear Authors,

Currently, we all know that most of the drugs used in the treatment of AD, only treat the symptoms, and the sole treatment option of Aduhelm is under the accelerated approval pathway. Thus we need drugs that are not only multi-target but privileged structures like the peptides that the authors have tested in this study.

This is systematic work, with clearly defined objectives. The authors have successfully achieved their objectives by careful design, execution, and interpretation of experiments. It is wonderful to know that the authors have filed for patents and that they are going to conduct pre-clinical trials in their future work.

A question I have is,

Although the authors explained that the RV deletion peptide variants performed consistently worse than the corresponding undeleted variants, for which they have provided experimental results. In figure 5C, why is the binding negative in this deleted variant?, even the BSA seems to be almost on the negative side.

My suggestions,

Figure 1

The use of arrows that specifically denotes inhibition could emphasize the figure and make it visually clear. Interaction, inhibition, and aggregation could be illustrated instead of representing these proteins as blocks.

Methods

In some instances, authors have typed the numbers in words instead of the number itself. For example, one instead 1. It would be better if this is consistent as numbers (like 1) everywhere.

Should/shouldn’t “in vitro” be italicized?

Author Response

We would like to thank this reviewer for his/her positive comments on this manuscript, and particularly for mentioning the need for multitarget and privileged structures to address the huge need for AD treatments. Please find the answers to your specific questions/suggestions below:

Question 1: Although the authors explained that the RV deletion peptide variants performed consistently worse than the corresponding undeleted variants, for which they have provided experimental results. In figure 5C, why is the binding negative in this deleted variant?, even the BSA seems to be almost on the negative side.

The values shown in Figure 5 are not the direct OD reading at 492 nm. They are the OD values of the wells containing the antigen (TLR4/MD2 in Figure 5C) minus the OD values of the corresponding wells containing no antigen (background signal obtained from non-specific binding of detection antibodies to the coated wells). When both OD values are very low, sometimes the background value is slightly higher than the corresponding value obtained in the presence of antigen. This explains why some of the values are slightly negative in Figure 5C.

Suggestion 1:

Figure 1

The use of arrows that specifically denotes inhibition could emphasize the figure and make it visually clear. Interaction, inhibition, and aggregation could be illustrated instead of representing these proteins as blocks.

We have greatly improved Figure 1 by following your suggestions.

Suggestion 2:

Methods

In some instances, authors have typed the numbers in words instead of the number itself. For example, one instead 1. It would be better if this is consistent as numbers (like 1) everywhere.

We have made this change to consistently have numbers written as numbers (and not letters) within the Methods section.

Suggestion 3:

Should/shouldn’t “in vitro” be italicized?

We have italicized “in vitro” and “in vivo” in the manuscript.